# The feasibility of a web-based resilience-building program to prevent stress among danish pregnant nulliparous women: A randomised controlled feasibility trial

Monica Ladekarl[1,2]*, Ina Olmer Specht[1,3], Amanda Rodrigues Amorim Adegboye[4,5], Anne Brødsgaard[2,6,7,8], Ellen Aagaard Nøhr[9], Nanna Julie Olsen[1], Berit Lilienthal Heitmann[1,3,10]

1 Research Unit for Diet and Health, the Parker Institute, Bispebjerg and Frederiksberg Hospital, Copenhagen, Denmark, 2 Department of Gynaecology and Obstetrics, Copenhagen University Hospital Amager Hvidovre, Copenhagen, Denmark, 3 Department of Public Health, Section for General Practice, University of Copenhagen, Copenhagen, Denmark, 4 Centre for Healthcare Research, Coventry University, Coventry, United Kingdom, 5 Centre for Agroecology, Water and Resilience, Coventry University, Coventry, United Kingdom, 6 Department of Paediatrics and Adolescent Medicine, Copenhagen University Hospital Amager Hvidovre, Copenhagen, Denmark, 7 Department of Public Health - Nursing and Health Care, Aarhus University, Aarhus, Denmark, 8 Department of People and Technology, Roskilde University, Roskilde, Denmark, 9 Department of Clinical Research, Research Unit for Gynaecology and Obstetrics, University of Southern Denmark, Odense, Denmark, 10 The Boden Group, Faculty of Medicine and Health, Sydney University, Sydney, Australia

* monica.ladekarl@regionh.dk

## Abstract

Stress during pregnancy has been found to modify aspects of fetal development, including organ maturation and neurodevelopmental problems, and to increase the risk of preterm birth and lower birth weight. Studies have shown lower stress levels and cortisol release in people with high resilience. To date, no randomized controlled trial (RCT) has evaluated the effect of a resilience-building digital program to prevent and cope with stress in pregnancy. This study assessed the feasibility of conducting a web-based resilience-building RCT among otherwise healthy pregnant women. In total, 124 nulliparous women were included at their first antenatal appointment in gestational weeks 14–20. The women were randomly allocated to the intervention group consisting of a web-based resilience-building program lasting 20 weeks or to a control group receiving usual care. This study evaluated recruitment, attrition, adherence, satisfaction, and compliance rates using questionnaires on adherence and satisfaction and qualitative phone interviews with pregnant women to investigate the acceptability and feasibility of the intervention. The adherence rate was 85%. The overall satisfaction with participation in the study was six on an 8-point Likert Scale, ranging from "terrible" to "fantastic". In the intervention group, 58% had a "good" or "very good" experience with the program, and 62% reported still using knowledge and methods in the program two months postpartum. At follow-up, 57% had

**Data availability statement:** The data set contains sensitive and potentially identifying information about participants and cannot be made publicly available due to ethical and legal restrictions. In accordance with Danish legislation and the requirements of the Danish Data Protection Agency (www.datatilsynet.dk), public sharing of de-identified data is not permissible, as full anonymization cannot be guaranteed. These restrictions were imposed by the Regional Committee on Health Research Ethics. However, data access can be granted to qualified researchers upon reasonable request and with appropriate approvals. Requests should be submitted to the study's principal investigator, Professor Berit L. Heitmann (Berit.Lilienthal.Heitmann@regionh.dk), or to the Research Unit for Diet and Health at The Parker Institute (bfh-eek@regionh.dk), which serves as the data access body.

**Funding:** Funded studies The study were supported by grants from: TrygFonden: https://www.tryghed.dk/ ID:125690. The Aase & Ejnar Danielsen Foundation: https://danielsensfond.dk/ 19-10-0094. The Dagmar Marshall Foundation: https://www.marshallsfond.dk/ No grant number. The Danish Association of Midwives: https://jordemoderforeningen.dk/ Jnr. 56-094/AMS The Parker Institute (funded by a core grant from the Oak Foundation): https://oakfnd.org/OFIL-24-074". The funders had no role in study design, data collection and analysis, decision to publish, or preparation of the manuscript."

**Competing interests:** The authors have declared that no competing interests exist.

**Abbreviations:** BMI, Body Mass Index; CD-RISC, The Connor-Davidson Resilience Scale; DASS, Depression, Anxiety, Stress Scale; FFQ, Food Frequency Questionnaire; GA, Gestational age; GWG, Gestational weight gain; PPWR, Postpartum weight retention; IQR, Interquartile Ranges; PSS, Perceived Stress Scale; RCT, Randomised Controlled Trial; RFQ, Reflective Functioning Questionnaire; SD, Standard Deviation.

completed all four questionnaires. Some women described that using resilience exercises during pregnancy and birth was helpful for their overall well-being. However, some women found receiving email reminders of the exercises stressful. Overall, our feasibility study showed that the intervention was feasible and acceptable. Before conducting a full-scale trial, minor modifications to the program and the delivery can be considered to improve acceptability and response rate. The trial was registered in the clinicaltrials.gov database (ID NCT03854331) on 26/02/2019 and approved by the Danish Data Protection (J.nr. VD-2019-13) and the National Committee on Health Research Ethics - Capital Region (J.nr. H-19000990).

# Background

## Introduction

Pregnancy is a vulnerable period for both the mother and the fetus, with stress being linked to numerous adverse outcomes. For example, studies have found that stress during pregnancy may increase the risk of preterm birth, reduce birth weight, and affect aspects of fetal development, including organ maturity and neurodevelopment [1–6]. For expectant mothers, stress may also lead to excessive weight gain during pregnancy and difficulties in losing postpartum weight [7].

There is no accurate estimate of the global prevalence of maternal stress during pregnancy due to the different screening methods used across different populations. However, studies suggest that many women of childbearing age experience stress in general and during pregnancy [8]. For example, figures from the Danish National Health Profile in 2021 showed that approximately 43% of Danish women between the ages of 16 and 44 years had feelings of stress according to the Perceived Stress Scale (PSS), using a cut-off of 18 points [9]. However, no previous studies have reported the occurrence or perception of stress in pregnant Danish women.

Stress is often defined as an individual perception that environmental demands exceed adaptive capacity [10] but can be conceptualized and measured in several ways [11]. Perceived stress is the subjective feeling of inner tension, while stressors are the acute or chronic stress exposures people experience (divorce, natural catastrophes, bereavement, unfavorable work conditions, or financial problems). The body's physiological reaction to stress, i.e., the level of stress hormones and their harmful physical consequences, are other ways to measure stress [12]. Finally, individual resilience or general coping ability also contributes to the individual's experience of stress [13].

Studies have shown reduced stress and cortisol release in people with high resilience [14,15]. Resilience is the successful adaptation to adversities, including successful recovery from adverse life events and sustainability in relation to life challenges [16]. Thus, building resilience may help reduce chronic stress by optimizing personal resources and tools to prevent the harmful effects of future exposure to stressors [17].

Among adults, improved resilience to stress has been shown to be an important promoter of adherence to sustained behavior change. Furthermore, resilience or

coping abilities during pregnancy may beneficially impact birth outcomes by attenuating the adverse behavioral, cognitive and physiological responses to stressors [18].

With the high prevalence of women feeling stressed, scalable solutions to help cope are needed. Studies indicate that digital interventions targeting mental health can be as effective as face-to-face interactions [19–21]. Internet-based interventions, e-health or m-health technologies, can be delivered by computer, tablet, or smartphone, and treatment can cost-effectively be accessed anytime and anywhere [22]. Nevertheless, evidence for the beneficial effects of exclusively digital interventions to prevent stress is still scarce for pregnant women [22]. In a systematic review conducted by Sakamoto et al. on the effects of mHealth psychosocial interventions on the psychosocial health of pregnant women, perceived stress was evaluated as an outcome in seven of the 44 included studies. Among these 7 studies, three reported positive effects, two presented mixed findings, and two found no significant effects. However, the evidence across all studies was rated as very low quality [23]. Although the maternal-fetal dyad is especially vulnerable to stress during pregnancy, little is known about whether the dyad is also receptive to salutary, resilience-promoting influences, and, as far as we know, digital resilience-building interventions to prevent stress have not yet been conducted in a pregnant population.

## Objectives

The current study assessed the feasibility of a randomized controlled trial (RCT) of a digital intervention to build resilience and prevent stress among nulliparous pregnant women. The research question was: What is the feasibility of conducting a RCT using a resilience-building program compared with standard care to prevent and cope with stress among pregnant women? Feasibility outcomes included 1) Feasibility of the recruitment strategy. 2) Acceptance of the study design, data collection and intervention. 3) Compliance, satisfaction, and retention rates. 4) Differences in attrition rate across age groups and other sociodemographic factors. 5) Key barriers and potential improvements needed before large-scale implementation of a main RCT. The CONSORT extension statement checklist for randomized pilot and feasibility trials was used as a guideline for reporting this study [24].

## Methods

### Study design

The study was a controlled single-center randomized feasibility trial with a parallel process evaluation. It compared a digital resilience-building program (MyResilience), detailed in the following method section, as an add-on to standard care to standard care alone.

### Setting

We recruited 124 first-time pregnant women from Copenhagen University Hospital, Amager-Hvidovre, in the Capital Region of Denmark, between June 2019 and November 2019. The obstetric department has approximately 7,000 births a year. Access to universal health care in Denmark is free, and all pregnant women are offered public antenatal care delivered by hospital-employed midwives, their general practitioners, and obstetricians or other specialists if needed. Women attending antenatal care at the hospital are healthy women, as well as women with medical and obstetric complications. However, in this study, we only recruited nulliparous pregnant women with a BMI < 30 kg/m2 receiving standard antenatal midwifery care. Pregnant women referred to specialist care because of past or present mental diagnoses, substance abuse, or psychosocial vulnerabilities were hence not eligible for recruitment [25].

### Randomization procedures

Pregnant women were randomized and stratified into two groups by midwifery center site (Amager and Hvidovre Hospitals) before baseline data collection in a 1:1 ratio to either intervention or control group using permuted block sizes. The

randomization was performed electronically via the browser-based Research Electronic Data Capture system (REDCap) (www.projectredcap.org). The randomization sequence was computer-generated by an external statistician not involved in the project. Due to the nature of the intervention, we could not blind the women to their allocation group.

### Ethics

The feasibility RCT was approved by the Danish Data Protection (J.nr. VD-2019-13) and the National Committee on Health Research Ethics - Capital Region (J.nr. H-19000990) and was registered in the clinicaltrials.gov database (ID NCT03854331) on 26/02/2019.

### Eligibility

Pregnant women eligible for this study were nulliparous, up to 20 weeks gestation, Danish speaking, not diagnosed with Type I or Type II diabetes, ≥ 18 and < 50 years, BMI < 30 kg/m$^2$ and singleton pregnancy.

### Sample size

Since this was a feasibility study, a formal power calculation for sample size was not performed. However, we aimed to recruit 120 participants, even though this is a relatively large number for a feasibility study [26]. This would allow us to examine the challenges when different healthcare personnel at a large delivery ward in a real-world situation, collect data and biological samples.

### Recruitment

According to the Danish Health Care Act, the research team was not allowed to screen the individual electronic journals for eligibility. Therefore, information about the study was distributed in three different ways: 1) All nulliparous women with a BMI < 30 kg/m$^2$ were sent information about the study to their personal and secure digital mailbox "e-Boks" linked to their unique personal identification number [27]. Subsequently, 2) A short oral presentation was given at the pregnancy classes, the voluntary pregnancy information meetings offered by the hospital to nulliparous women. Furthermore, 3) the research team was present in the waiting room at the midwifery center where the antenatal clinical examinations took place. The women self-selected for the study by contacting the research team. All women who expressed interest in participating in the study were evaluated against the inclusion/exclusion criteria and enrolled after written consent. Participant flow through the study process is depicted in the flow diagram (Fig 2).

### Intervention design

The intervention was a digital adaption of the MyResilience program developed earlier by the Danish Committee for Health Education [28] and specifically tailored to pregnant women by instructors from The Danish Committee for Health Education in collaboration with our research group. First, we conducted three focus group interviews examining stress and worries in healthy pregnant women [25]. Then, three research group members attended a three-day training course that introduced them to the regular use of the program. Finally, based on the results of the focus group interviews, the instructors from The Danish Committee for Health Education selected 20 modules from their extensive collection of more than 70 exercises and information texts. The program was delivered via a webpage that the women could access via phone, tablet, or computer and was based on mentalization, mindfulness, exercises to improve self-control, and cognitive behaviour therapy [28]. Six modules included easily understood information about mentalization, the function of the brain, the consequences of how we think about ourselves and others, and the concepts of socialization and attention. The other 14 modules were practical exercises. Examples of exercises were: Reflexive questions about own thoughts, relaxation techniques, breathing techniques, writing down positive everyday life experiences each night before bedtime, writing down

how you think other people perceive you and how you perceive yourself, trying different ways to calm your brain when it is in a state of alarm, and making and prioritizing your to-do list. The women in the intervention group could choose between accessing the web page's modules as text or listening to an audio file. They were verbally instructed on how to use the MyResilience program by the research team and subsequently prompted to use it by a weekly email with a link to a specific module on the project website. They could also access all modules immediately after randomization if preferred. The reminder encouraged them to read and/or listen to the information and to do the exercises provided. All pregnant women in the intervention group were further invited to participate in a two-hour lecture elaborating on the concept of resilience, given by the Danish Committee for Health Education.

## Feasibility outcomes

**Acceptability of the study design and data collection.** We assessed the suitability of the study design, how the randomization and data collection procedures worked, and whether the participating women considered the procedures appropriate and acceptable. The data came from a participant-log Excel spreadsheet that the research team filled out for every participant, from conducted field notes and phone interviews with the pregnant women. As a part of the feasibility related to data collection assessment, maternal hair was collected for cortisol analyses at baseline, gestational age (GA) 28, GA 35 and two months after birth. In addition, infants' hair was collected at birth and two months after birth. The participant's acceptance of donating the samples and the midwives' willingness to collect the samples were recorded.

**Recruitment strategy.** We calculated the number of participating pregnant women relative to the predicted number of eligible pregnant women to assess the recruitment rate and suitability of the eligibility criteria. We also assessed the number of days and working hours used to recruit 120 women. The data came from our participant-log Excel spreadsheet and the local hospital database.

**Attrition.** We recorded the number of pregnant women in each study arm dropping out from the study when they dropped out and the reasons for dropping out. The women were asked why they wished to stop, and the reason was recorded in the participant-log Excel spreadsheet.

**Adherence.** To assess adherence, we recorded the degree to which the behaviour of the participating women corresponded to the intervention assigned to them. The use of the resilience program was measured from web statistics, self-reports in the questionnaires, and qualitative open-ended evaluation interviews by phone.

**Intervention and study satisfaction.** A questionnaire using an 8-point Likert-type scale ranging from "terrible" to "fantastic" and qualitative open-ended phone interviews assessed the overall satisfaction with the study and the intervention.

## Primary mental health outcome

Our primary assessment outcome was the experience of stress measured by Cohen's Perceived Stress Scale (PSS). PSS measures the respondent's experience of stress over the past four weeks using ten questions on to what extent life is experienced as unpredictable, uncontrollable, and stressful and whether the respondent feels nervous or stressed. The scale goes from 0 to 40, with higher scores indicating higher stress levels [29]: Scores of 0–13 would be considered low stress, scores of 14–26 as moderate stress, and scores of 27–40 as high stress [30].

## Secondary mental health outcomes

We used the Connor-Davidson Resilience Scale (CD-RISC) to assess the participant's resilience [31]. The scale comprises 25 items, each rated on a 5-point scale (0–4). It reflects the self-reported ability to tolerate unpleasant experiences such as changes, personal burdens, pressure, failure and pain [32]. The total score ranges from 0-100, with higher scores reflecting greater perceived resilience [31]. We used the Depression Anxiety Stress Scales (DASS), a 42-item instrument

designed to measure the three related negative emotional states of depression, anxiety, and stress. The scale goes from 0 to 42 for every emotional state. The higher the score, the higher the level of depression, anxiety, or stress experienced [33]. To assess the participants' mentalization ability, we applied The Reflective Functioning Questionnaire (RFQ), comprising eight items and two subscales [34]. Lower scores represent an acknowledgement of the opacity of mental states, a characteristic of good mentalization [35]. Fear of childbirth and fetal health anxiety were evaluated by asking: "Are you anxious about the course of the upcoming delivery/the health of the expected child?". Possible responses were: "Not at all", "A little", or "A lot". Only the last response is considered to represent severe fear of childbirth or fetal health anxiety [36,37].

## Other explorative outcomes

While the study's primary purpose was to test feasibility, the research team also collected various demographic information and health outcome data considered relevant for the main study, particularly weight-related outcomes. The information also included, among others, stressful life events during pregnancy (i.e., divorce, losing their job, bereavement), physical activity, diet, smoking and alcohol use, sleep quality and quantity, general health, method of birth, and pregnancy complications.

## Data sources

Questionnaire data were collected electronically at four time points: Just after randomization at GA 14–20 (baseline), GA 28, GA 35 and two months postpartum (Fig 1). At GA 14–20 (baseline), the participants also received a separate validated 360-item Food Frequency Questionnaire (FFQ) covering diet in the past four weeks [38]. All in all, they received five questionnaires during the study period. Information about prepregnancy BMI, gestational age at birth, birth weight, and birth-related outcomes was obtained from electronic medical records.

## Qualitative process evaluation

A question guide was developed based on data compiled from the questionnaires to evaluate the intervention's form, content, and execution. The guide included questions such as "Which exercises do you remember?" and "What were your impressions of the website and the reminders?", aiming to explore participants' experiences in more depth. The purpose of the evaluation was to understand the perception of the intervention better, as the participants could not elaborate very much in the questionnaires. The evaluation of the intervention was initially planned to be performed in face-to-face focus group interviews with the intervention group during the spring of 2020, shortly after the completion of the intervention. However, focus group interviews were cancelled due to the Danish Covid-19 lockdown, and telephone interviews were conducted from October to December 2020 instead.

## Statistical analysis

The sociodemographic and general participant characteristics were described using medians and interquartile ranges (IQR) or means and standard deviations (SD) for continuous variables and absolute numbers (n) and percentages (%) for categorical variables. Comparisons between groups at baseline were made using Student's t-test for normally distributed

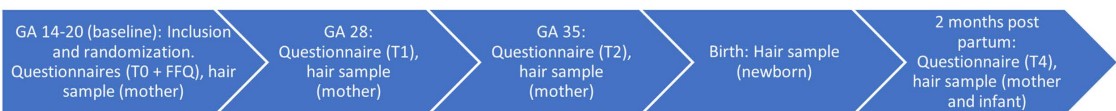

**Fig 1. Timeline for the study.**

continuous variables and the Kruskal-Wallis test for non-normal samples. We used Fisher's exact test for dichotomous variables and Pearsons's chi2 test for non-ordered categorical variables. Descriptive statistics were used for feasibility outcomes.

As this was a feasibility study, no formal hypothesis testing was conducted. However, to explore potential trends, we conducted simple linear regression analyses to estimate adjusted mean differences with 95% confidence intervals between groups. These were adjusted for baseline values of the corresponding outcome measures. The aim was not to infer effectiveness but to inform the design and sample size estimation for a future RCT.

Statistical analyses were performed using SAS Enterprise Guide® software, version 7.15 (SAS Institute Inc., Cary, NC, USA). Alpha levels < 0.05 were considered statistically significant.

### Qualitative analysis

The data from telephone interviews used to evaluate the intervention was divided into themes. We used a deductive content analysis approach based on Elo & Kyngäs [39], applying a predefined coding matrix focusing on form, content, and execution. Two researchers coded the interview transcripts and resolved discrepancies through discussion. The emerging themes and categories were mapped to the predefined matrix. When doing so, only aspects that matched the matrix were chosen from the data [39]. Quotes from participants were used to illustrate the themes.

## Results

Flowchart of enrolment, allocation, and follow-up of study participants (Fig 2).

### Recruitment

The number of women informed about the study reflects "broad feasibility-focused recruitment" via oral information and distribution of leaflets in antenatal clinics and pregnancy classes. Inclusion criteria were clearly stated in the leaflet, and potential participants self-referred only if they believed themselves eligible. This approach, combined with the absence of electronic pre-screening, resulted in low exclusion numbers post-contact. After receiving written and verbal information, a total of 124 women out of 1054 women contacted the project staff, wishing to participate in the study. Of these, 61 women (of 658) were recruited at the pregnancy classes, and 63 women (of 396) were recruited from the waiting rooms in the antenatal clinics. The recruitment rate was five women/week (124 randomised women recruited in 23 weeks). The recruitment rate was higher in the pregnancy classes (1 participant/one h) compared to the midwifery centres (1 participant/6 h).

### Participants

The mean age was 30.3 ± 4.0 years, ranging from 22 to 43 years. Ethnicity was 97% Danish or European descent. Most women reported living with a spouse or partner (97%, n = 110) and had completed medium or high education (82%, n = 93).

At baseline, according to their Perceived Stress Scale (PSS) scores, 37.7% reported moderate stress, and 62.3% had low stress scores. The mean PSS score was 11.8 out of 40. The mean score was 71.9 out of 100 on the Connor-Davisson resilience scale. The average DASS scores were all within the normal range. A total of 13,4% had severe childbirth fear, and 24,1% had severe fetal health anxiety at baseline.

### Randomization

The randomization process worked well with an unpredictable allocation group assignment without any statistical imbalances (p < 0.05) in demographic data or mental health outcomes at baseline (Table 1).

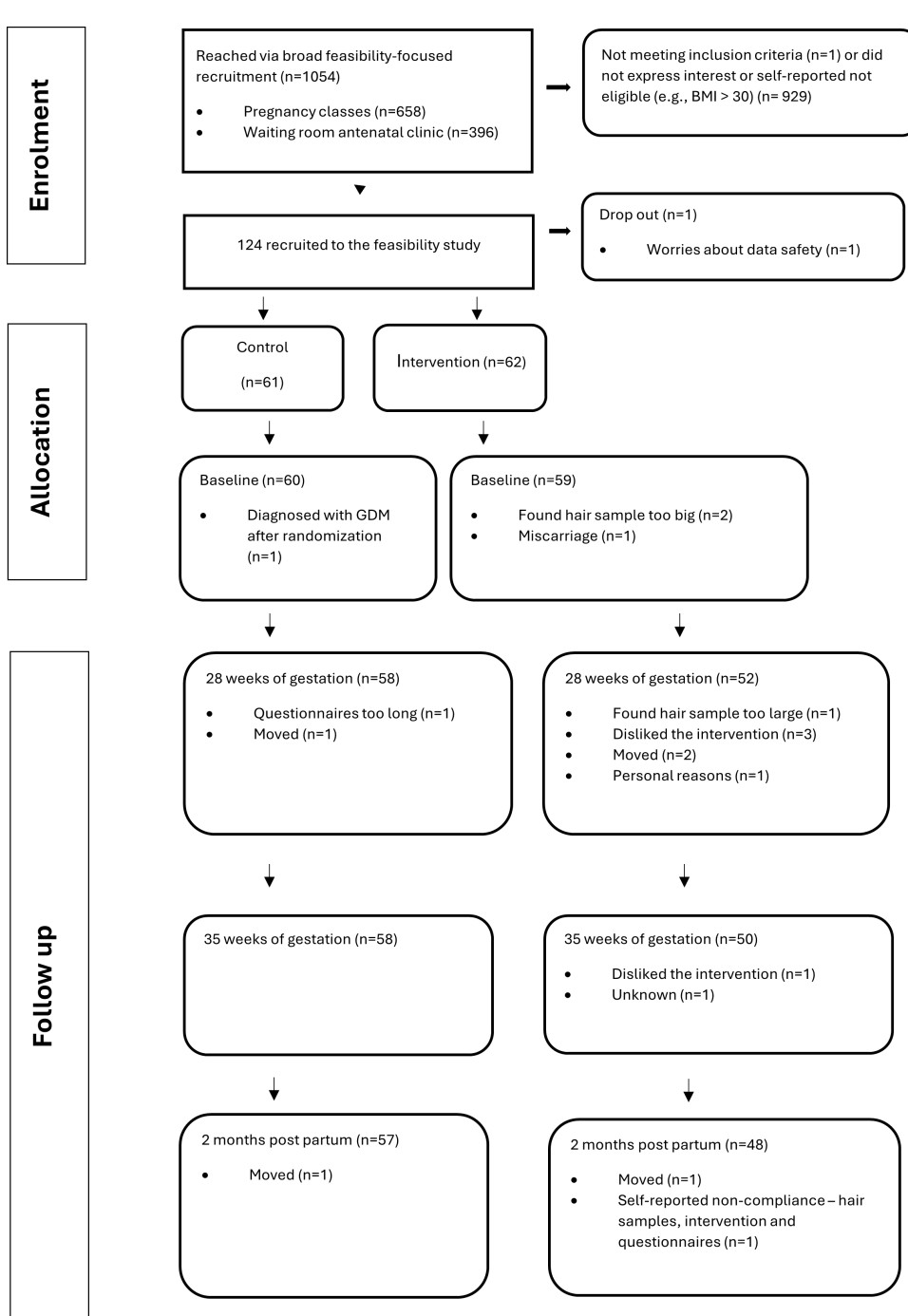

**Fig 2. Flowchart of enrolment, allocation, and follow-up of study participants.**

**Table 1. Baseline maternal characteristics.**

| Variables at baseline | Intervention n=62 | | Control n=62 | |
|---|---|---|---|---|
| | Median (IQR) or n (%) | n missing | Median (IQR) or n (%) | n missing |
| Age (years) | 30 (5.00) | 1 | 29 (4.50) | 2 |
| Gestational age (weeks) | 18.0 (2.3) | 1 | 17.6 (1.6) | 2 |
| Prepregnancy BMI (kg/m2) | 22.0 (3.7) | 1 | 22.0 (3.9) | 2 |
| PSS (score) | 11 (7) | 5 | 12 (7) | 5 |
| CD-RISC (score) | 71 (11.00) | 6 | 69 (11.50) | 6 |
| DASS depression (score) | 1 (3) | 5 | 2 (3) | 5 |
| DASS anxiety (score) | 2 (4) | 5 | 2 (3) | 5 |
| DASS stress (score) | 6 (6) | 5 | 6 (9) | 5 |
| RFQc (score) | 0 (0.33) | 6 | 0 (0.33) | 5 |
| RFQu (score) | 1.50 (1.42) | 6 | 1.33 (1.17) | 5 |
| Fetal health anxiety (yes) | 15 (25) | 6 | 12 (20) | 6 |
| Birth anxiety (yes) | 7 (11) | 6 | 8 (13) | 6 |
| Highest obtained education (years) | | 5 | | 5 |
| • High school or less | 3 (5) | | 8 (14) | |
| • Short (up to 3 years) | 6 (11) | | 4 (7) | |
| • Medium (3–5 years) | 15 (26) | | 23 (40) | |
| • High (> 5 years) | 33 (58) | | 22 (39) | |

IQR: Interquartile range. PSS: Perceived Stress Scale. A higher score (max. 40) indicates higher perceived stress. CD-RISC: The Connor-Davidson Resilience Scale. A higher score (max. 100) indicates higher resilience. DASS: Depression, Anxiety, Stress Scale. Higher scores (max. 28 each) indicate greater severity of stress, anxiety and depression. RFQ: Reflective Functioning Questionnaire. Certainty (RFQc) and uncertainty (RFQu) about mental states. The score can range from 0-3. A lower score indicates higher mentalisation ability.

## Attrition

There were 19 (15%) women dropping out during the study period, 14 (23%) in the intervention arm, 4 (7%) in the control arm, and one before randomization. Demographic data for the latter was subsequently deleted at the participant's demand. By mistake, the data of two other drop-outs were also deleted from our database (one from each group). The most frequent primary cause for drop-out was that the participant relocated to another city during the study (n=5), disliked the intervention (n=4), or found the size of the donated hair sample too large (n=3). See other reasons in Fig 3.

The retention rate at GA 28 was 89% (110 of 124 randomised women). At the end of the study, the retention rate was 85% (105 of the 124 randomised women).

There was no statistical difference in baseline characteristics between drop-outs, participants with complete questionnaire data, and participants without complete data, except for the DASS depression score, which was higher in the non-completers (P=.05) (Table 2).

## Questionnaire compliance

At enrolment, 90% (112/124) filled out the first questionnaire sent by email immediately after inclusion and randomization, while 75/124 (60%) at the same time also completed the FFQ. Of the initially included women, 43% (53/124) completed all five questionnaires, while 50% (53/105) of those retained at follow-up completed all five questionnaires. Without considering the FFQ, 57% (71/124) and 68% (71/105), respectively, finished all four questionnaires. The response rate was higher in the control group than in the intervention group, except at baseline (Table 3). For example, among participating women

PLOS Mental Health

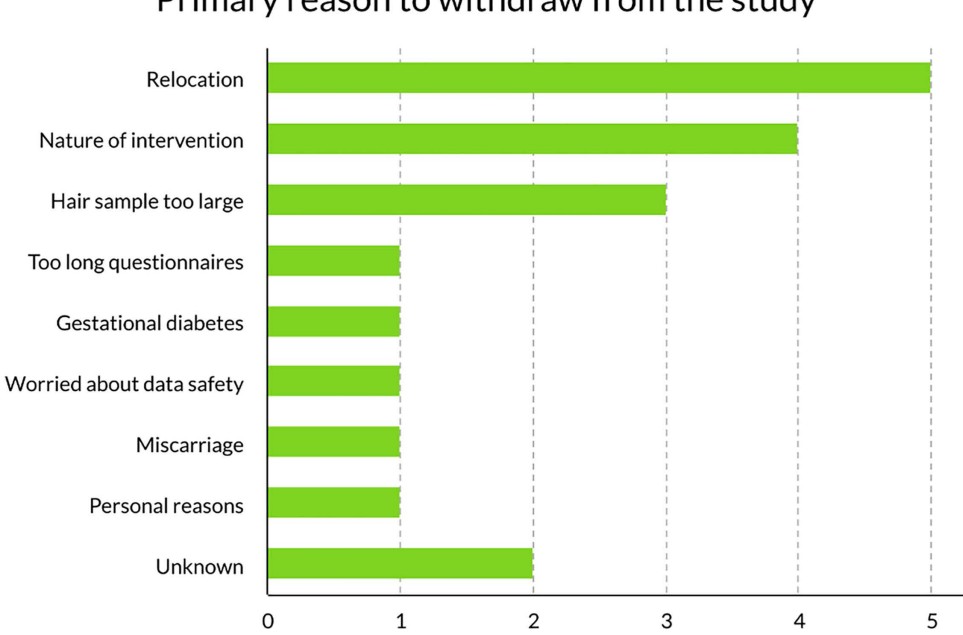

**Fig 3. Primary reason to withdraw from the study.**

in GA 35, the completion rate was 74% in the intervention group vs. 86% in the control group, but this was not significant (*P* = .14).

## Hair sample compliance

During pregnancy, approximately 80% of the hair samples were taken by the hospital midwives at GA 28 and 35. At birth, 57% 71/124 newborn hair samples were taken. The primary reasons listed for not collecting the hair sample were too little hair (20%), the woman leaving the hospital before the sample could be collected (17%), and the newborn being admitted to the Neonatal Intensive Care Unit immediately after birth (14%).

## Adherence to the intervention

The response rate for questions related to adherence to the intervention was 67% of the women remaining in the intervention group 2 months postpartum (n = 32/48). Of these, 16% (6/32) reported that they had visited the project home page with the MyResilience intervention program 1–2 times per week, and 84% (27/32) less than once per week. It was not possible to answer "never visited the project homepage". The most common reason given for not using the MyResilience homepage more than weekly was lack of time. For technical reasons, further interpretation of the complementary web statistics was not possible as this data, related to GDPR regulations, was not collected at the individual level. The reported barriers to using the program can be seen in Fig 4.

Table 4 presents exploratory adjusted mean differences between intervention and control groups in gestational week 35, based on simple linear regression analyses controlling for baseline values of the corresponding outcomes. For example, the adjusted mean difference in Perceived Stress Scale (PSS) score was –3.26 (95% CI: –5.55; –0.96), indicating a statistically significant and potentially meaningful reduction in perceived stress in the intervention group. Similarly, the

**Table 2. Baseline characteristics of the participants with complete questionnaire data, 0-—4 questionnaires completed, and drop-outs.**

| Variables at baseline | Complete (4 questionnaires+FFQ) | | Non-complete (0–4 questionnaires) | | Dropouts | |
|---|---|---|---|---|---|---|
| | n=53 | | n=52 | | n=19 | |
| | Median (IQR) or n (%) | n missing | Median (IQR) or n (%) | n missing | Median (IQR) or n (%) | n missing |
| Age (years) | 30.0 (4.0) | 0 | 29.5 (5.50) | 0 | 31.0 (7.0) | 3 |
| Gestational age at inclusion (weeks) | 17.9 (1.6) | 0 | 17.5 (2.4) | 0 | 18.3 (1.1) | 3 |
| Prepregnancy BMI (kg/m²) | 23.1 (3.9) | 0 | 21.8 (3.2) | 0 | 23.2 (5.0) | 3 |
| PSS (score) | 12.0 (7.0) | 0 | 12.0 (7.0) | 3 | 8.5 (8.0) | 7 |
| CD-RISC (score) | 70.0 (12.0) | 0 | 70.0 (8.0) | 5 | 74.0 (17.5) | 7 |
| DASS depression (score) | 1.0 (2.0) | 0 | 2.0 (3.0) | 3 | 1.0 (6.0) | 7 |
| DASS anxiety (score) | 2.0 (3.0) | 0 | 3.0 (4.0) | 3 | 2.5 (5.0) | 7 |
| DASS stress (score) | 6.0 (6.0) | 0 | 6.0 (7.0) | 3 | 7.0 (8.0) | 7 |
| RFQc (score) | 0.0 (0.3) | 0 | 0.1 (0.3) | 4 | 0.0 (0.1) | 7 |
| RFQu (score) | 1.5 (1.3) | 0 | 1.3 (1.1) | 4 | 1.6 (1.6) | 7 |
| Fetus health anxiety (yes) | 11 (21) | 0 | 12 (26) | 5 | 4 (33) | 7 |
| Birth anxiety (yes) | 9 (17) | 0 | 5 (11) | 5 | 1 (8) | 7 |
| Highest obtained education (years) | | 0 | | 3 | | 7 |
| High school or less | 3 (6) | | 6 (12) | | 2 (17) | |
| Short (up to 3 years) | 3 (6) | | 6 (12) | | 1 (8) | |
| Medium (3–5 years) | 17 (32) | | 17 (35) | | 4 (33) | |
| High (>5 years) | 30 (57) | | 20 (41) | | 5 (42) | |

IQR: Interquartile range. PSS: Perceived Stress Scale. A higher score (max. 40) indicates higher perceived stress. CD-RISC: The Connor-Davidson Resilience Scale. A higher score (max. 100) indicates higher resilience. DASS: Depression, Anxiety, Stress Scale. Higher scores (max. 28 each) indicate greater severity of stress, anxiety, and depression. RFQ: Reflective Functioning Questionnaire. Certainty (RFQc) and uncertainty (RFQu) about mental states. The score can range from 0-3. A lower score indicates higher mentalization ability.

**Table 3. Questionnaire compliance for 124 participants.**

| Time point | Questionnaires completed | | | | | |
|---|---|---|---|---|---|---|
| | Retained participants | | | All initially included participants | | |
| | All n (complete)/ n (retained) (%) | Intervention n (complete)/ n (retained) (%) | Control n (complete)/ n (retained) (%) | All n (%) | Intervention n (%) | Control n (%) |
| Baseline (14–20 weeks) | 112/120 (93) | 56/59 (95) | 56/61 (92) | 112 (90) | 56 (90) | 56 (90) |
| FFQ (14–20 weeks) | 75/120 (67) | 41/59 (70) | 34/61 (56) | 75 (60) | 41 (66) | 34 (55) |
| 28 weeks | 96/110 (87) | 45/52 (87) | 51/58 (88) | 96 (77) | 45 (73) | 51 (82) |
| 35 weeks | 87/108 (81) | 37/50 (74) | 50/58 (86) | 87 (70) | 37 (60) | 50 (81) |
| 2 months pp | 81/105 (77) | 36/48 (75) | 45/57 (79) | 81 (65) | 36 (58) | 45 (73) |

FFQ: Food Frequency Questionnaire.

PP: Postpartum.

adjusted difference in resilience (CD-RISC) was+3.39 (95% CI: 0.64; 6.14), indicating a potential improvement in participants' ability to cope with stress.

Due to the small sample size and the feasibility design, these findings should be interpreted with caution. However, the results suggest favorable trends across several mental health outcomes, including perceived stress, depression, and

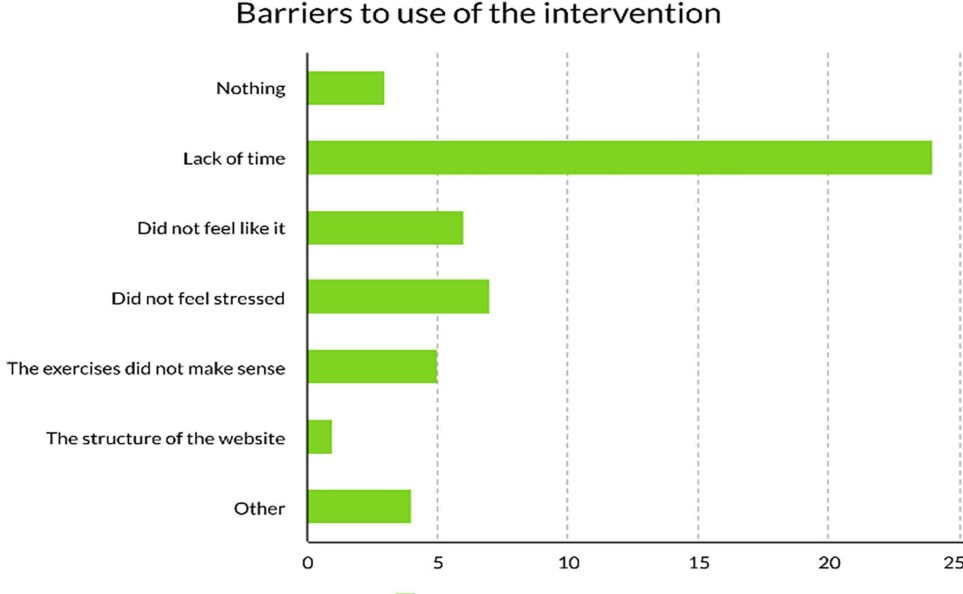

**Fig 4. Barriers to use of the intervention.**

**Table 4. Indication of intervention effects.**

| Mental Health Outcomes GA 35 | n | Intervention | Control | Mean diff. (95% CI) | Adj. mean diff. (95% CI)* |
|---|---|---|---|---|---|
| PSS (score) | 87 | 9.32 | 13.38 | -4.06 (-6.70; 1.41) | -3.26 (-5.55; -0.96) |
| DASS depression (score) | 87 | 1.73 | 2.71 | -1.59 (-3.38; 0.20) | -1.13 (-1.92; 0.65) |
| DASS anxiety (score) | 87 | 2.30 | 3.86 | -1.56 (-3.2; 0.10) | -0.77 (-2.20; 0.67) |
| DASS stress (score) | 87 | 5.49 | 8.30 | -2.81 (-5.42; -0.20) | -1.41 (-3.49; 0.66) |
| CD-RISC (score) | 87 | 73.97 | 71.18 | 2.79 (-1.16; 6.74) | 3.39 (0.64; 6.14) |
| Fetal health anxiety (yes) | 87 | 11 | 22 | -11 (-26,02; 4,04) | n/a |
| Fear of Childbirth (yes) | 87 | 0 | 20 | -20 (-31,09; -8,91) | n/a |
| **Maternal weight outcomes** | **n** | **Intervention** | **Control** | **Mean diff. (95% CI)** | **Adj. mean diff. (95% CI)*** |
| GWG (kg) | 87 | 13.05 | 13.45 | 0.41 (-2.01; 1.89) | -0.42 (-2.02; 1.19) |
| PPWR (kg) | 79 | 2.89 | 4.42 | -1.54 (-3.23; 0.15) | -1.59 (-3.29; 0.11) |
| **Neonatal outcomes** | **n** | **Intervention** | **Control** | **Mean diff. (95% CI)** | **Adj. mean diff. (95% CI)*** |
| Birth weight (g) | 110 | 3435 | 3463 | -28 (-211; 156) | n/a |
| Gestational age birth (days) | 110 | 281.44 | 281.22 | 0.22 (-3.52; 3.96) | n/a |

*Adjusted for baseline value.

GA: gestational age. PSS: Perceived Stress Scale. A higher score (max. 40) indicates higher perceived stress. DASS: Depression, Anxiety, Stress Scale. Higher scores (max. 28 each) indicate greater severity of stress, anxiety and depression. CD-RISC: The Connor-Davidson Resilience Scale. A higher score (max. 100) indicates higher resilience. GWG: Gestational Weight Gain. Change in self-reported weight from prepregnancy to 35 weeks (kg). PPWR: Postpartum Weight Retention. Change in self-reported weight from prepregnancy to 2 months pp (kg).

anxiety, although confidence intervals often included the null. For maternal weight-related outcomes, postpartum weight retention (PPWR) favored the intervention group with an adjusted difference of –1.59 kg (95% CI: –3.29; 0.11), but were not statistically significant, while no effects were observed for birth weight or gestational age. At the two-month postpartum follow-up, no group differences were observed.

## Satisfaction with the study and the intervention

The median overall satisfaction with the study was 6 (out of 8), where 1 was "terrible" and 8 was "fantastic". The satisfaction was significantly higher in the control group compared to the intervention group, 6.6 vs 5.5 ($P = .002$). In the intervention group, 58% (17/29) of the women had a "good" or "very good" experience with the program, and 62% (18/29) reported still using the knowledge, methods, or tools two months after giving birth. The response rate for these questions was 60% of the women retained at 2 months postpartum.

## Qualitative findings

All women from the intervention group were contacted after the last questionnaire, and 37 interviews were completed. The qualitative data are presented in Table 5.

The women generally found the MyResilience website and program user-friendly and easy to navigate. Most participants mentioned the same 2–3 modules when asked what they remembered the best: A story about coping with unwelcome thoughts and two different breathing exercises. These modules were described in a positive context. All three modules were among the first seven exercises that the women were introduced to. Most women reported reading the assignments whenever they received a reminder email but did not always do the exercises. Some pooled reminder emails together and looked at several tasks at a time. The intervention was also considered stressful for some and was described as a "chore" and like doing "homework". At the same time, others felt that the exercises did not apply to them because they did not feel stressed. Some said they already knew many of the exercises via birth preparation classes, work arrangements, or previous courses and felt they were already "saturated", and did not think they had gained new knowledge or tools. It made these women feel guilty whenever they received a reminder email when they had not done the exercises they were supposed to do. Some requested more guidance in the audio files so the exercise could be performed while listening to the audio file and not subsequently. As a result, they could not remember the exercise when they had finished the audio file.

**Table 5. Overview of qualitative findings.**

| Themes | Categories | Quotes |
|---|---|---|
| Helpful | • Helpful before and during the birth<br>• Awareness of not being stressed<br>• Being able to feel your body through breathing | *"When I read it, I thought 'it is very elementary', but it actually worked during pregnancy … I was nervous because we have been in treatment for infertility, and I have lost [an unborn child] before."*<br>*"Yes, that square, the breathing square. I used it a lot, also during birth. It was good. It gave me something to focus on to handle the pain better."*<br>*"Yes, I can remember some of them. There was one I mainly used and that I still use now. When an unwelcome thought comes, you have to think a little about it, notice whether it is uncomfortable, and say "Hello hello" [to the unwelcome thought]. I think it was a good exercise. I still think about it now if I have too many thoughts, if I need to relax, or if some thoughts take up to much space. Then I can think a little about the thought and say "Goodbye".* |
| It feels like chores or homework | • The extent of the study<br>• Quantity of emails | *"It was a bit of a stressor; it spoke to my bad conscience; I had an unread email in my inbox all the time. I saved it as Unread so that I would remember it. Nevertheless, I felt I was falling behind all the time."*<br>*"And because I had so much else to do, it kind of became a remember-thing... I set up a folder in my emails because I had to put them aside because it became too stressful."* |
| Wrong casting | • No general need<br>• Assumptions about worries<br>• Feelings of obligation | *"The thing with assuming that I was worried – I did not like that very much."*<br>*"It was as if I had been put in a category that I had a problem in the pregnancy, like one that had mental difficulties, and I did not. So for me, I did not belong in that group. I did not have that need."*<br>*"Did it mainly for your sake. I do not feel that I fit in, personality-wise."* |
| Already experienced | • Other birth preparation<br>• Work-related experience<br>• Earlier courses or training | *"I also went to private birth preparation [sessions], where we also did breathing [exercises], and that was what I prioritized."*<br>*"I also went to birth preparational course, [there was] a lot of the breathing exercises, and mentalizing, so I think that there were many exercises that were similar."* |

When asked if they preferred an app to a website, the women had mixed opinions. Some found that an app would have been more easily accessible, while others pointed out that app notifications quickly disappear from your phone compared to emails that could be saved in the inbox. In addition, some women expressed a wish to perform exercises "on the go", i.e., in a car or train. However, it was difficult because the audio files were not "guiding", and the website was less easy to navigate on cell phones. When asked about preferences related to the program's content, they preferred the more practical or physical exercises to the modules only containing educational information. For example, some women described using the relaxing breathing exercises during pregnancy and birth as helpful.

## Discussion

Psychological stress during pregnancy is a significant public health concern due to its potential adverse effects on maternal well-being and birth outcomes. Despite increasing interest in digital mental health support, evidence for stand-alone digital interventions designed to prevent stress or build resilience in pregnant women remains limited [22]. Our study addresses this gap by evaluating the feasibility and preliminary impact of *MyResilience*, a web-based resilience program adapted for pregnant women.

Although the study was not powered to assess effectiveness, our exploratory analyses suggested modest improvements in perceived stress and resilience among participants. This aligns with previous research showing that digital tools teaching mindfulness, acceptance, and cognitive coping strategies can improve emotional regulation and reduce stress, both generally and during pregnancy - including potential downstream benefits for infant outcomes [18,21,40–43]. Several participants reported using breathing exercises to manage daily stressors or sleep problems and highlighted the value of learning to shift unhelpful thoughts—skills that are central to resilience-building. Taken together, our findings suggest that the intervention may hold promise in helping pregnant women prevent or better manage stress, though this remains to be confirmed in a larger trial. While the modest reductions in perceived stress and increases in resilience were exploratory, they are noteworthy considering the relatively low intensity of the intervention and the high baseline resilience in our sample. However, the mixed user experience also underscores that perceived relevance and timing of such interventions are critical to their effectiveness. For example, participants who reported benefiting from the tools often described using them proactively in everyday stress situations, suggesting that the intervention may have supported their ability to cope with stressors as they arose—core to the concept of resilience. In contrast, women who did not perceive themselves as stressed were less likely to engage. Future studies should more closely investigate how individual differences in baseline stress, coping styles, and readiness to engage affect the uptake and effect of resilience-building interventions during pregnancy.

RCTs are considered the gold standard for evaluating the effect of a particular intervention. However, their effectiveness depends on the participants' adherence, retention and acceptance of the intervention [44]. Therefore, conducting feasibility studies before a large-scale RCT is launched is essential to understand what influences these factors, the resources needed to conduct the study or implement the intervention, what number of eligible participants to include, identify potential risks and challenges that may arise during the project, and examine the time required to collect and analyze data [24,45].

Enrolment of an adequate number of participants to test the study protocol was achieved. Participants were recruited over a period of 5 months, and the five women/week recruitment rate was considered acceptable. However, calculations based on the local hospital database suggested that as many as 30 women per week may have been eligible. Therefore, in a large-scale RCT, adjusting the recruitment strategy would be beneficial to ensure that recruitment targets are met in time and that study completion is not delayed. The recruitment rate might have improved by involving the hospital midwives in the project at the inclusion stage. Having multiple recruitment strategies to ensure the reach of the target may be beneficial. However, the challenge is that it may take many additional resources, so using the most effective or cost-effective is essential.

Even though women were not considered eligible if they received specialized antenatal care because of psychosocial problems, 37.7% still self-reported moderate stress at baseline according to the PSS scores, which corresponds well with the proportion of women of childbearing age reporting feelings of stress in the Danish National Health Profile [46]. In addition, we found that 13% had severe childbirth fear, and 24% had severe fetal health anxiety at baseline. These numbers slightly decreased throughout pregnancy (to 11% and 17% at gestation week 35). Severe childbirth fear was almost twice as high as the 7.6% found among nulliparas from a large Danish cohort study from 2008 (37).

Still, the participants' resilience and mentalization ability scores were already relatively high at baseline, suggesting that many were already functioning well psychologically, potentially making improvements in resilience and mentalization scores more challenging as we were using a relatively low-dose intervention in this target group. This ceiling effect may partly explain the modest overall gains and underscores the importance of refining the target group and the intervention.

However, making the intervention more intensive would lower scalability and increase expenses. Another potential enhancement could involve expanding the scope of the full-scale trial to encompass all participants, including those with pre-existing stress diagnoses, and assessing whether this subgroup may benefit most from the intervention [47]. While this would reduce generalizability, it might improve the ability to detect meaningful change.

Almost 70% of women answered all four main questionnaires. Although this can be considered a high response rate, all questionnaires should undergo additional revision to try to optimize the response rate further. For instance, some women commented that the questionnaires were too long. In addition, the FFQ could not be completed using a cell phone, and it was completed to a lesser extent than the other questionnaires. Delivering the program via an app (including data collection tools/ forms) may be expected to improve compliance and participant experience. More women in the control group (86%) than the intervention group (74%) answered the questionnaires at the end of the intervention period (GA 35), and more control group women came to the two-month postpartum visit (85% vs. 67%), implying a potential "study fatigue" in the intervention group. On the other hand, only 4/14 women in the intervention group reported that intervention dissatisfaction was the main reason for dropping out.

A higher number of drop-outs in the intervention group compared to the control group has also been found in other similar studies, suggesting that it can be challenging to retain participants in an intervention group [48–50]. Drop out in the intervention group may also, to some extent, be due to chance; for instance, only women in the intervention group found the collected hair samples too large.

However, we cannot rule out whether the participant burden in this study could have played a role. For example, in addition to the questionnaires, the women in the intervention group were sent 20 emails in 20 weeks asking them to complete the modules. In the qualitative interviews, some women reported that the volume of emails could create stress, which indicates that the content and the number of emails must be re-evaluated before the full-scale trial. About 60% of the intervention group participants reported they had a "good" or "very good" experience with MyResilience. However, about 40% reported a very poor or below average experience, and none reported an excellent experience. This may be related to the fact that some women did not feel stressed and, consequently, may not have thought they were the target group for MyResilience. On the other hand, other women felt stressed and could probably have benefited from the exercises but did not have the energy or time to do them. Two women reported a very poor experience: One had a below-average PSS score (5 points), and the other had an above average PSS score (15 points). It is not surprising that a specific intervention does not appeal to everyone. Some people prefer one-to-one therapy to group treatment, while others prefer digital interventions to face-to-face. A certain rate of drop-outs and non-completers must be expected even in the best-designed intervention, but tailoring the recruitment screening process and the intervention to increase participant engagement in the future RCT will be necessary [51].

The program was adapted to pregnant women but had not previously been used in this group. Neither had it been used as a stand-alone digital intervention, but it had been used, e.g., by teachers and their students. We did three focus group interviews before adapting the intervention. Before rolling out a full-scale trial, we will further involve pregnant women in

the program adaptation process to help highlight specific challenges and priorities that may not be immediately apparent and ensuring that the intervention is directly relevant and applicable to the needs, circumstances, and challenges of pregnant women.

Our study has several limitations. First, a future RCT would benefit from better measurement of the actual use of the intervention on an individual level, which we did not do in this study. The initial plan was to track the individual participants' use of the intervention using a technical solution, but this data could not be obtained due to GDPR issues (web page hosted on a server outside Europe). However, it will be possible to access this information in future studies by using a different technical setup and a European cloud. Additionally, the web statistics provided by the Danish Committee for Health Education could not be interpreted for this study. As a result, compliance had to be assessed using self-reported questionnaires and phone interviews, and the exact frequency of participant use of the intervention website is uncertain. Consequently, it was not possible to evaluate the impact of usage on study satisfaction or intervention effectiveness.

In addition, enrolment at only two hospital sites may limit the generalisability of our results. On the other hand, more than 7000 women or 1 in 8 women living in Denmark, give birth annually at the Amager-Hvidovre Hospitals.

Due to the nature of the intervention, blinding of participants and study staff was not possible, which may have introduced expectancy or performance bias, particularly for self-reported outcomes such as perceived stress. This is a known challenge in behavioral intervention trials, where active blinding is often not possible [52]. However, except for standardised reminder emails and the option to attend a one-time information session, there was no direct contact with the intervention group during the study period, limiting the risk of differential attention bias. To decrease bias in future studies, we will consider blinding outcome assessors and employing an active comparator to balance expectancy effects across groups.

Focus group interviews were planned face-to-face but were done using telephone interviews due to the Danish COVID-19 lockdown. Although we do not anticipate this to be a major limitation, some issues may have been lost using less personal and intimate interviews by phone.

Strengths of the study include comparing different recruitment methods and using both quantitative and qualitative data sources to assess the feasibility and acceptability of recruitment, data collection, and intervention. Also, our primary mental health measures, namely PSS and CD-RISC, were previously validated in Danish [53,54]. Finally, the scaling to 124 pregnant women made it possible to do a more robust quantitative assessment of feasibility items and, not least, allowed us to try different solutions to the challenges when different healthcare personnel at a large delivery ward collect data and biological samples.

However, it is important to underline that our feasibility study was not designed or powered to test a hypothesis that will be tested in a future RCT. Therefore, we cannot conclude about the intervention's effect on stress or resilience outcomes. Although the study was not powered to assess effectiveness, the exploratory adjusted estimates may still inform future sample size calculations and provide preliminary insights into possible effect sizes.

## Future perspectives

Building psychological resilience to manage stress may offer a novel and important potential avenue to limit stress and stress-related consequences in pregnant women. However, further research is necessary to develop this area thoroughly. Furthermore, as evidenced in prior feasibility trials, ensuring that participants acquire and properly utilize the skills provided through e-health interventions can be challenging. To address this challenge and ensure effective outcomes, additional research is needed to investigate the delivery and uptake of these interventions across multiple devices and platforms.

Utilizing e-health technologies presents an attractive, low-cost solution for intervention delivery. However, more extensive trials are required to determine their effectiveness [55] in full-scaled randomized and controlled interventions to estimate effects not only on stress alleviation or resilience building but potentially also on pregnancy and birth outcomes to develop this area further.

## Conclusion

Overall, this feasibility study showed that moving on to a full-scale RCT of a home-based stress-preventing and resilience-building digital intervention seems feasible and that the intervention will be acceptable and well tolerated by participants. However, some changes to the program will be needed before the rollout of the full-scale trial. Based on the findings of the feasibility study, we propose the following specific changes before launching a full-scale RCT: (1) Reducing questionnaire burden and adapting all formats for mobile use; (2) Minimizing perceived pressure by reducing reminder frequency and allowing customization; (3) Using GDPR-compliant platforms to track individual usage and improve data quality; (4) Involving pregnant women directly in tailoring the intervention to ensure relevance and feasibility; and (5) broaden the target group to also include pregnant women with known stress.

## Acknowledgments

Thank you to the participating pregnant women, the research assistants, the Danish Committee for Health Education and the Hvidovre Hospital's delivery ward staff.

## Author contributions

**Conceptualization:** Monica Ladekarl, Ina Olmer Specht, Nanna Julie Olsen, Berit Lilienthal Heitmann.

**Data curation:** Monica Ladekarl, Ina Olmer Specht.

**Formal analysis:** Monica Ladekarl, Ina Olmer Specht.

**Funding acquisition:** Monica Ladekarl, Ina Olmer Specht, Nanna Julie Olsen, Berit Lilienthal Heitmann.

**Investigation:** Monica Ladekarl.

**Methodology:** Monica Ladekarl, Ina Olmer Specht, Amanda Rodrigues Amorim Adegboye, Anne Brødsgaard, Nanna Julie Olsen, Berit Lilienthal Heitmann.

**Project administration:** Monica Ladekarl.

**Supervision:** Ina Olmer Specht, Amanda Rodrigues Amorim Adegboye, Anne Brødsgaard, Ellen Aagaard Nøhr, Nanna Julie Olsen, Berit Lilienthal Heitmann.

**Visualization:** Monica Ladekarl.

**Writing – original draft:** Monica Ladekarl.

**Writing – review & editing:** Monica Ladekarl, Ina Olmer Specht, Amanda Rodrigues Amorim Adegboye, Anne Brødsgaard, Ellen Aagaard Nøhr, Nanna Julie Olsen, Berit Lilienthal Heitmann.

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
