## [Decision Letter · Decision Letter 0]

4 Apr 2025

PMEN-D-25-00028

The Feasibility of a Web-based Resilience-building Program to Prevent Stress among Danish Pregnant Nulliparous Women: A Randomised Controlled Feasibility Trial

PLOS Mental Health

Dear Dr. Ladekarl,

Thank you for submitting your manuscript to PLOS Mental Health. After careful consideration, we feel that it has merit but does not fully meet PLOS Mental Health’s publication criteria as it currently stands. Therefore, we invite you to submit a revised version of the manuscript that addresses the points raised during the review process.

We look forward to receiving your revised manuscript.

Kind regards,

Hanif Abdul Rahman, Ph.D.

Academic Editor

PLOS Mental Health

Journal Requirements:

https://journals.plos.org/mentalhealth/s/figures

https://journals.plos.org/mentalhealth/s/figures#loc-file-requirements

2. In the online submission form, you indicated that “By request”.

a. In a public repository,

b. Within the manuscript itself, or

c. Uploaded as supplementary information.

Additional Editor Comments (if provided):

Reviewers' comments:

Reviewer's Responses to Questions

**Comments to the Author**

1. Does this manuscript meet PLOS Mental Health’s publication criteria?

Reviewer #1: Yes

Reviewer #2: Yes

2. Has the statistical analysis been performed appropriately and rigorously?

Reviewer #1: Yes

Reviewer #2: Yes

3. Have the authors made all data underlying the findings in their manuscript fully available (please refer to the Data Availability Statement at the start of the manuscript PDF file)?

Reviewer #1: No

Reviewer #2: Yes

4. Is the manuscript presented in an intelligible fashion and written in standard English?

Reviewer #1: Yes

Reviewer #2: Yes

Reviewer #1: The current paper describes a feasibility/pilot trial for web-based Resilience to prevent stress among pregnant nulliparous women in the Danish population. The study has many strong points,

1. Being a feasibility study no power analysis is attempted, rather sample size is based on feasibility

2. The results correctly do not include any statistical test or p-values

3. The emphasis is on the feasibility of recruitment, randomization, retention, etc.

I find the results reported have confidence intervals which is also appropriate.

Some more clarification could be helpful, such as

1. As per the CONSORT diagram, the difference between screening and randomization is unclear. It seems only one subject is lost between the two phases. If true this is remarkable and needs more discussion.

2. It is stated, in conclusion, that some changes are needed in the large-scale RCT based on what is learned from the feasibility trial. This must be stated more elaborately as this is the primary reason for conducting any feasibility trial.

3. Since the trial can not be blinded to the subject its implication on the subject-level bias needs further attention.

4. While statistical testing is not needed some aspect of statistical modeling will be helpful, which is completely missing from the manuscript.

Reviewer #2: The Feasibility of a Web-based Resilience-building Program to Prevent Stress among Danish Pregnant Nulliparous Women: A Randomised Controlled Feasibility Trial

PMEN-D-25-00028

This was a very interesting and well written article. The study looked at the Feasibility of a Web-based Resilience-building Program to Prevent Stress among Danish Pregnant Nulliparous Women. The title of the article matches the information discussed in the article. The abstract and introduction of the study shows that authors have thought this research through. They also provided enough detail relating to their methods and the decisions surrounding their chosen methods.

Even though their article was well written, there are potential areas of improvement. The first area that I would encourage authors to strengthen is their discussion. The discussion felt like a repeat of the methods section or something that should be fused with the methods section. It did not discuss the second part of your topic (ie preventing stress) but was more on the technicality of the findings. I was looking forward to hearing authors’ debates and perspectives on the ‘did the program build pregnant women’s resilience to stress’, especially because authors presented their findings very well. Situating the discussion within existing research would also benefit the study. Please also add more detail on how you analysed your telephone interviews. Provide more information on the data analysis method used and the decisions you took to eventually get to your themes. Your conclusion can also be improved too so that the article ends with the same intensity that you introduced it with.

Otherwise, I enjoyed reading this article.

**Do you want your identity to be public for this peer review?** For information about this choice, including consent withdrawal, please see our Privacy Policy

Reviewer #1: No

Reviewer #2: **Yes: ** Dr Nombuso Gama

---

## [Editor Report · Decision Letter 1]

12 Jul 2025

The Feasibility of a Web-based Resilience-building Program to Prevent Stress among Danish Pregnant Nulliparous Women: A Randomised Controlled Feasibility Trial

PMEN-D-25-00028R1

Dear Mrs. Ladekarl,

We are pleased to inform you that your manuscript 'The Feasibility of a Web-based Resilience-building Program to Prevent Stress among Danish Pregnant Nulliparous Women: A Randomised Controlled Feasibility Trial' has been provisionally accepted for publication in PLOS Mental Health.

Best regards,

Hanif Abdul Rahman, Ph.D.

Academic Editor

PLOS Mental Health